# An End-to-End Underwater-Image-Enhancement Framework Based on Fractional Integral Retinex and Unsupervised Autoencoder

**Yang Yu** 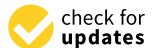 **and Chenfeng Qin ***

School of Marine Science and Technology, Northwestern Polytechnical University, Xi'an 710068, China
* Correspondence: chenfeng_qin@mail.nwpu.edu.cn

**Abstract:** As an essential low-level computer vision task for remotely operated underwater robots and unmanned underwater vehicles to detect and understand the underwater environment, underwater image enhancement is facing challenges of light scattering, absorption, and distortion. Instead of using a specific underwater imaging model to mitigate the degradation of underwater images, we propose an end-to-end underwater-image-enhancement framework that combines fractional integral-based Retinex and an encoder–decoder network. The proposed variant of Retinex aims to alleviate haze and color distortion in the input image while preserving edges to a large extent by utilizing a modified fractional integral filter. The encoder–decoder network with channel-wise attention modules trained in an unsupervised manner to overcome the lack of paired underwater image datasets is designed to refine the output of the Retinex. Our framework was evaluated under qualitative and quantitative metrics on several public underwater image datasets and yielded satisfactory enhancement results on the evaluation set.

**Keywords:** underwater image enhancement; fractional integral Retinex; unsupervised autoencoder

## 1. Introduction

About 70% of the earth's surface is covered by water, so exploring and utilizing marine resources benefits humankind greatly. However, vision-based underwater detection tasks face many challenges caused by the poor quality of underwater images. On the one hand, underwater images always present awful visibility due to the scattering and absorption of light while it propagates in the water. On the other hand, the more rapid attenuation of red light than the green and blue light in the water results in severe distortion of color rendition in digital images.

Similar to the physical dehazing model [1,2] on land, there are some classical underwater image restoration algorithms based on physical models, for example, methods based on the Jaffe–McGlamery underwater imaging model [3,4]. Sea-thru [5] is a representative algorithm that combines the physical model and the principle of digital image processing. Researchers from the Univ. of Haifa developed a model that takes an RGB-D image as input, estimates backscatter in a way inspired by the dark channel prior, and uses an optimization framework to obtain the range-dependent attenuation coefficient. However, some models based on the physical assumption of underwater imaging rely on extra devices, such as the Laser Underwater Camera Imaging Enhancer [6] and Coulter Counter [7], to acquire specific parameters, and even the Sea-thru requires an additional depth sensor. Thus, computer-vision algorithms that are independent of expensive measurements are preferred for general underwater image enhancement tasks.

First introduced by Land and McCann [8], the Retinex theory has been extensively applied in multiple research areas such as dehazing, enhancement of remote sensing images, and underwater images. One of the most noted variants of Retinex, called Multi-Scale Reinex [9], captured light changes under different scales and removed the light from the

input image to simultaneously achieve dynamic range compression, color consistency, and lightness rendition. A color restoration function based on empirical parameters was applied to the output after Retinex processing to recover authentic color from the degraded image. As most Retinex algorithms employ Gaussian operators to estimate the light image, note that underwater images are generally used for close-range detection of the reef, shellfish, ancient ruins, and underwater vehicles, and sharp edges naturally exist in them. Hence, the Gaussian filtering operator used by classical MSR algorithms may cause over-smoothing and a lack of details. According to Qi Wang et al. [10], fractional calculus operators excel at processing information with weak derivatives that the texture structure contains. More specifically, the fractional differential operator has the characteristics of memorizing and a higher signal-to-noise ratio than the integer differential operator, which enables the fractional operator to capture more detailed information. For the fractional integral operator, it attenuates the high-frequency portion of the signal dramatically while enhancing the low part to a large extent. The denoising method based on fractional integral was first proposed by Huang et al. [11], who constructed an eight-direction fractional integral operator to filter input noise from the original image directionally. We noticed that most of the fractional integral-based filters were limited to a $3 \times 3$ or $5 \times 5$ size, which was not compatible with the Retinex theory that the light image should be estimated by a much larger filter to capture a trend of slow change of the light. In the Multi-Scale Retinex, a three-scale parallel Retinex network always uses 15, 80, and 250 as sigma values for the Gaussian filter to predict the light, leading to filters of size $45 \times 45$, $240 \times 240$, $750 \times 750$, respectively. For such a considerable filter size, the fractional method could either expand the filter kernel size or carry out recurrent filtering. However, the former will not work because the filter became sparse and resulted in a "fringe phenomenon" on the image, while the latter required significant time cost.

With the development of the use of deep learning theory in image processing, approaches based on convolutional neural networks have shown potential for image enhancement tasks. For underwater image dehazing, J. Perez et al. introduced a model based on classical CNN architecture [12]. This work utilized a dataset of pairs of raw and restored images for training a network so that clear images could be restored from degraded inputs. Based on the multi-branch design, the UIE-Net proposed by Yang Wang et al. [13] firstly extracts features from the input image using a sharing network and then uses two subnets—the color correction network (CC-Net) and the haze removal network (HR-Net)—to simultaneously achieving color correction and haze removal. Taking into account that pretraining is necessary for these supervised deep learning-based approaches, a problem occurred, namely that there were limited underwater image datasets available for training the networks. To overcome the lack of paired underwater images, an amount of paired datasets for image enhancement tasks were proposed, for example, the UIEB dataset [14] proposed by Li et al. Containing 950 authentic underwater images under various light conditions with 890 paired images in them, the UIEB dataset uses classical enhancement methods and artificial selection to generate reference underwater images. Although paired image-based datasets boost supervised methods, we should be aware that acquiring absolute ground truth for underwater image enhancement tasks is practically impossible. In fact, since a significant portion of the paired datasets were synthetic, models trained on these datasets yielded undesirable generalization performance and sometimes failed on authentic underwater images.

To surmount the challenges, methods based on unsupervised learning could be a possible solution. One direction for unsupervised underwater image enhancement is to utilize depth-guided networks, represented by the WaterGAN [15]. While the first part of the WaterGAN aims to generate underwater images from the in-air ones, the second part restores underwater images by successively passing the input through a depth estimation network and a color correction network. Taking an underwater image and corresponding output relative depth map of the depth estimation network as input, the color-correction network restores the color of the input image. The other direction is based on pure computer

vision, independent of any specific underwater imaging model. Inspired by unsupervised methods on low-light image enhancement, image translation, and style transfer, Lu et al. proposed an adaptive algorithm with multi-scale cycle GAN and dark channel prior (MCycleGAN) [16]. By using the dark channel prior to obtain the transmission map of an image and designing an adaptive loss function to improve underwater image quality, the raw images being multi-scale calculated were able to convert to clear results. The TACL [17] proposed by Risheng Liu et al. utilized a bilateral constrained closed-loop adversarial enhancement module to preserve more informative features and embedded a task-aware feedback module in the enhancement process to narrow the gap between visually-oriented and detection-favorable target images.

Rather than the above-mentioned unsupervised methods, which mostly used GAN to generate enhancement results or classical underwater image restoration algorithms based on specific physical models, in this paper, by combining the Retinex algorithm and unsupervised image enhancement approaches, we propose an end-to-end underwater-image-enhancement framework which generates the enhanced image primarily using the Retinex. The underwater image pre-enhanced by the Retinex will be further improved by an encoder–decoder network trained in an unsupervised style to yield better contrast and luminance enhancement, as well as more satisfactory perceptual performance.

The main contributions of this paper are as follows:

- A fractional integral-based Retinex and an improved fractional-order integral operator, which eliminated the drawback of classical fractional integral operator for large-kernel filtering and resulted in more accurate estimation for light images, was proposed in this paper.
- An effective unsupervised encoder–decoder network requiring no adversarial training and yielding perceptually pleasing results was employed to refine the output of the previous Retinex model.
- Combining the fractional integral-based Retinex and unsupervised autoencoder mentioned above, the proposed end-to-end framework for underwater image enhancement was evaluated on several public datasets and produced impressive results.

The rest of this paper is organized as follows: In Section 2, the mathematical background of fractional double integral, on which the proposed variant of Retinex is based, is briefly introduced. In Section 3, we first propose the FDIF-MSR algorithm and then illustrate an end-to-end framework based on FDIF-MSR and the unsupervised encoder–decoder network. In Section 4, the proposed image enhancement model is evaluated on three public datasets, and some of the results are shown in this paper. In Section 5, the conclusion is given.

## 2. Mathematical Background

### 2.1. The Definition of Fractional Derivatives

There are many definitions of fractional-order derivatives. However, Riemann–Liouville (R-L), Grünwald–Letnikov (G-L), and Caputo gave the three most commonly used definitions of fractional derivatives. The Grünwald–Letnikov definition is deduced from the expression of integer-order differential, whereas the other two are derived from the integer-order integral Cauchy formula.

1. Let $\alpha$ be a positive real number. When $n - 1 \leqslant \alpha < n$, where $n$ is a positive integer, the left-hand Riemann–Liouville fractional derivatives can be written as:

$$_aD_t^\alpha f(t) = \frac{d^n}{dt^n}\left(\frac{1}{\Gamma(n-\alpha)}\int_a^t \frac{f(\tau)}{(t-\tau)^{\alpha-n+1}}\,dx\right) \tag{1}$$

where $\alpha$ is called the order of the R-L derivative.

2. The Grünwald–Letnikov definition of fractional derivatives is defined as

$$_aD_t^\alpha f(t) = \lim_{h \to 0} h^{-\alpha} \sum_{j=0}^{[(t-a)/h]} (-1)^j \binom{\alpha}{j} f(t-jh) \tag{2}$$

where $\binom{\alpha}{j}$ are the binomial coefficients, $[\cdot]$ denotes the integer part.

3. We have Caputo's definition of fractional derivatives, which is defined as:

$$_a^C D_t^\alpha f(t) = \frac{1}{\Gamma(n-\alpha)} \int_a^t \frac{f^{(n)}(\tau)}{(t-\tau)^{\alpha-n+1}} dx \tag{3}$$

There is an equivalent relation between Caputo and R-L derivatives such that for a positive real number $\alpha$, satisfied $0 \leqslant n-1 < \alpha < n$, if function $f(t)$ defined on the interval $[a, b]$ has continuous derivatives of $n-1$ order and integrable, we have

$$_aD_t^\alpha f(t) =_a^C D_t^\alpha f(t) + \sum_{j=0}^{n-1} \frac{f^{(j)}(a)(t-a)^{j-\alpha}}{\Gamma(1+j-\alpha)} \tag{4}$$

Similarly, R-L and G-L derivatives are also equivalent when $f(t)$ satisfies the above conditions. The G-L definition is the most commonly used for the numerical calculation of fractional derivatives. By extending the binomial coefficients to the field of real numbers, a more general form of the G-L derivatives can be written:

$$_aD_t^\alpha = \lim_{h \to 0} h^{-\alpha} \sum_{j=0}^{[(t-a)h]} (-1)^j \frac{\Gamma(1+\alpha)}{\Gamma(\alpha-j+1)\Gamma(j+1)} f(t-jh) \tag{5}$$

Note that all definitions of fractional derivatives above are left-handed; the corresponding right-hand derivative of the G-L definition is defined similarly by the expression

$$_aD_t^\alpha = \lim_{h \to 0} h^{-\alpha} \sum_{j=0}^{[(t-a)h]} (-1)^j \frac{\Gamma(1+\alpha)}{\Gamma(\alpha-j+1)\Gamma(j+1)} f(t+jh) \tag{6}$$

*2.2. Fractional Integral*

To extend fractional differential to integral, according to fractional operator theory, we use integral operator $I^{-1}$ instead of the differential operator $D$ and fractional order $-\nu$ instead of $\alpha$. For the G-L definition, the fractional integral of order $\nu$ can be written as:

$$_aI_t^\nu = \lim_{h \to 0} h^\nu \sum_{j=0}^{[(t-a)h]} (-1)^j \frac{\Gamma(1-\nu)}{\Gamma(1-\nu-j)\Gamma(j+1)} f(t-jh)$$
$$= \lim_{h \to 0} h^\nu \left( f(t) + \nu f(t-h) + \frac{\nu(\nu+1)}{0} f(t-2h) + \cdots \right) \tag{7}$$

Replace the coefficients of $f(t-jh)$ with $\omega_j^{(\nu)}$; the equation becomes

$$_aI_t^\nu = \lim_{h \to 0} h^\nu \sum_{j=0}^{[(t-a)h]} \omega_j^{(\nu)} f(t-jh) \tag{8}$$

while $\omega_j^{(\nu)}$ can be recurrently calculated by

$$\begin{cases} \omega_0^{(\nu)} = 1 \\ \omega_j^{(\nu)} = (1 - \frac{1-\nu}{j})\omega_{j-1}^{(\nu)} \end{cases} \tag{9}$$

### 2.3. Fractional Double Integral

The fractional integral can be enlarged for a double integral. Assuming that $x \in (a, c)$ and $y \in (b, d)$, for rectangular domain $(a, c) \times (b, d)$ in $\mathbb{R}^2$, consider $\alpha$-order fractional integral derived from the left-hand G-L definition:

$$_b I_y^\alpha (_a I_x^\alpha f(x, y)_{XL})_{YL} = \lim_{h_y \to 0} h^\alpha \sum_{k=0}^{[(y-b)/h_y]} \{\omega_k^{(\alpha)} [\lim_{h_x \to 0} h^\alpha \sum_{j=0}^{[(x-a)/h_x]} \omega_j^{(\alpha)} f(x - jh, y - kh)]\} \tag{10}$$

For casual signal $f(x, y)$, divide the interval $(a, c)$ and $(b, d)$ into equal parts using step $h_x = h_y = 1$, which also conforms to the fact that the sampling step on a 2-D image matrix is one unit; then, we can eliminate the limit sign and obtain

$$_b I_y^\alpha (_a I_x^\alpha f(x, y)_{XL})_{YL} = \sum_{k=0}^{[y-b]} \{\omega_k^{(\alpha)} [\sum_{j=0}^{[x-a]} \omega_j^{(\alpha)} f(x - jh, y - kh)]\} \tag{11}$$

where $\omega_j^{(\alpha)}$ and $\omega_k^{(\alpha)}$ can be calculated by Equation (9), respectively.

## 3. Proposed Method

In this section, we first put forward an improved fractional integral filter and then combine it with the MSRCR algorithm. Furthermore, an unsupervised encoder–decoder network is utilized to improve the quality of images processed by the Retinex. The pipeline of the proposed method is shown in Figure 1.

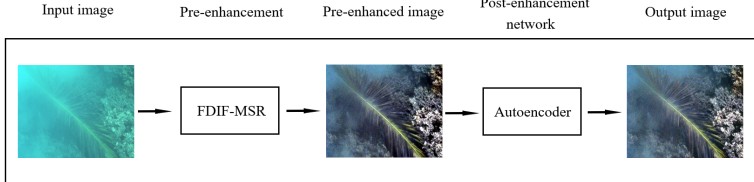

**Figure 1.** Diagram of the proposed end-to-end underwater-image-enhancement framework.

### 3.1. Fractional Double Integral Filter (FDIF)

A classical eight-direction fractional integral filter is constructed as shown in Figure 2. Recalling Equation (8), we can note that a more simplified version of left fractional single integral by replacing variable $h$ with unit one can be written as:

$$_a I_t^\alpha = \sum_{j=0}^{t-a} \omega_j^{(\alpha)} f(t - jh) \tag{12}$$

Generalizing this simplified equation to eight directions, involving directions along the positive and negative x-axis, positive and negative y -axis, and the 45°, 135°, 225°, and 315° directions along the positive x-axis in the counter-clockwise direction, we were able to construct the eight-direction fractional integral operator.

| $W_{fm}$ | 0 | 0 | $W_{fm}$ | 0 | 0 | $W_{fm}$ |
|---|---|---|---|---|---|---|
| 0 | $\ddots$ | 0 | $\vdots$ | 0 | $\iddots$ | 0 |
| 0 | 0 | $W_{f1}$ | $W_{f1}$ | $W_{f1}$ | 0 | 0 |
| $W_{fm}$ | $\cdots$ | $W_{f1}$ | $W_{f0}$ | $W_{f1}$ | $\cdots$ | $W_{fm}$ |
| 0 | 0 | $W_{f1}$ | $W_{f1}$ | $W_{f1}$ | 0 | 0 |
| 0 | $\iddots$ | 0 | $\vdots$ | 0 | $\ddots$ | 0 |
| $W_{fm}$ | 0 | 0 | $W_{fm}$ | 0 | 0 | $W_{fm}$ |

**Figure 2.** The classical 8-direction fractional integral filter on a 2-D grid.

The coefficients of the classical eight-direction filter are as follows:

$$\begin{cases} W_{f0} = 1 \\ W_{f1} = \frac{\nu(\nu+1)}{2} \\ W_{f2} = \frac{\nu(\nu+1)(\nu+2)}{6} \\ \cdots \\ W_{fm} = \frac{\nu(\nu+1)(\nu+2)\ldots(\nu+m-1)}{m!} \end{cases} \tag{13}$$

Specifically when $\nu = 1$, $W_{fi} = 1$ where $i = 0, 1, \ldots, m$.

As we mentioned above and show in Figure 3, such a filter presents a "fringe phenomenon" when the filter matrix size grows and becomes sparse.

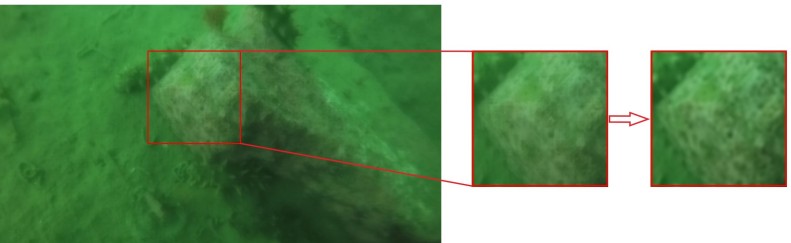

**Figure 3.** The fringe phenomenon caused by a sparse filter matrix. A sparse filter could only capture changes of pixels in the 8 directions; therefore starlike artifacts were found in the denoised image. The rightmost image patch shows the result of the FDI filter, and the artifacts were perfectly removed. We also made a quantitative comparison between the input image and filtered images: $SSIM(FI, input) = 0.939$ and $SSIM(FDI, input) = 0.947$. This image is from the RUIE dataset [18], and the SSIM can be calculated by Equation (23).

To address the issue, we replace the fractional integral filter by introducing a fractional double integral. According to Equation (11), the coefficients in fractional integral filter kernel are rearranged as in Figure 4. Considering the upper left part of the fractional filter deduced from the lefthand G-L definition of the fractional integral, it is obvious that along the x-axis and y-axis, coefficients are consistent with those used in the eight-direction filter, while they decay exponentially along the diagonal. Mathematically, the upper right part can be obtained by using the right-hand and left-hand fractional integral formula for $x$ and $y$, and the Equation (11) becomes

$$_b I_y^\alpha {}_a I_x^\alpha f(x,y)_{XR,YL} = \sum_{k=0}^{y-b} \omega_k^{(\alpha)} \sum_{j=0}^{x-a} \omega_j^{(\alpha)} f(x+jh, y-kh) \tag{14}$$

The other two parts can be obtained similarly. By using the additivity property of the convolution operation and assuming the four parts of the filter be $F_i, i = 1, 2, 3, 4$, four quadrants of integral can be calculated by one single kernel

$$F_{FDIF} = \sum_{i=1}^{4} F_i \tag{15}$$

A normalization operation is performed after the FDIF kernel is calculated to avoid introducing extra energy into the image matrix. A visualized version of the FDIF kernel is shown in Figure 4.

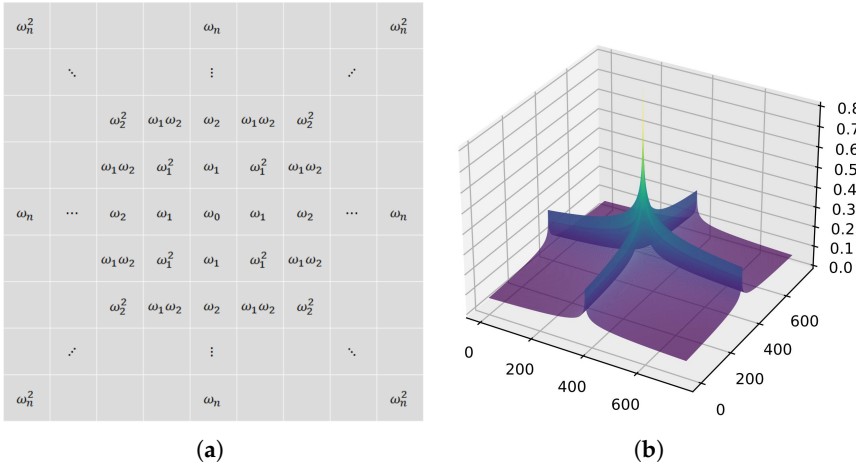

(a)          (b)

**Figure 4.** FDIF kernel based on the fractional double integral. (**a**): Coefficients in the kernel; (**b**): Visualized version of the kernel by Matplotlib [19]. Unlike in the 8-direction operator, all of the coefficients in the kernel are unequal to zero as long as $1 - \alpha \neq j$. Since we assume $\alpha > 0$ and $\omega_0^{(\alpha)} = 1$, the coefficients are all nonzero. In particular, the coefficients all become one when the fractional order $\alpha$ equals 1, and the filter becomes an average filter. Coincidentally, some of the Retinex implementations use average filtering in practice to estimate the light to lower computational costs.

### 3.2. Multi-Scale Retinex with FDIF

In this section, let us briefly recall the basic structure of the MSRCR algorithm and then optimize it with our FDIF filter.

#### 3.2.1. Retinex Theory

Briefly, the Retinex models the imaging process to show that objects in the image have particular reflection properties, and therefore, it can be found that incident light led to various color performances. The essential purpose of the Retinex model is to remove the influence of the light so that objects' characteristics can be recovered from the noised image. The primitive Retinex modeled the light as multiplicative noise such that

$$R(x, y) = I(x, y) \times L(x, y) \tag{16}$$

The single-scale Retinex process is given by

$$\log R(x, y) = \log I(x, y) - \log L(x, y) = \log I(x, y) - \log[F(x, y) * I(x, y)] \tag{17}$$

where $R(x, y)$ represents the enhanced image, $I(x, y)$ represents the original input image, and $L(x, y)$ denotes the light image, which is also considered as noised image, and the light image can be estimated by $F(x, y) * I(x, y)$. Most of the time, assuming the light changes

slowly, the Gaussian kernel consequently becomes an appropriate implementation of the light estimation operator. The Gaussian surround function is given by Jobson [9] as

$$F_{Gaussian}(x, y) = Ke^{-\frac{r^2}{c^2}} \tag{18}$$

where $c$ denotes Gaussian surround space constant, $r$ denotes the distance from current pixel $(x, y)$ to the center $(x_0, y_0)$, and $K$ is chosen so that the kernel brings in no extra gain. The MSR integrates Gaussian surround functions or, more generally, light estimation operators under various scales. The multi-scale calculation can be expressed by

$$R = \sum_{i=1}^{n} R_i \tag{19}$$

where $n$ denotes the number of scales, and the overall $R$ can be obtained by summation of images enhanced by different scales of a basic Retinex process.

In Retinex theory, a small-scale Gaussian kernel can achieve dynamic range compression, while a sizeable Gaussian kernel specializes in tonal and color rendition. A third intermediate scale combines both advantages of the small kernel and the large kernel; meanwhile, it eliminates the "halo" artifacts near sharp edges. The principle for choosing the Gaussian surround scale, proposed by Jobson [9], is that the sigma values of the Gaussian filter are 15, 80, and 250, corresponding to the three scales. According to the three-sigma rule, the filters' kernel size should be $45 \times 45$, $240 \times 240$, and $750 \times 750$.

### 3.2.2. Combination of FDIF and MSR

In this section, a modified MSRCR algorithm based on the fractional double integral filter is proposed. By directly superseding the Gaussian filter in the single-scale Retinex process, the FDIF operator estimates the light image while preserving edges in the original noised image. As shown in Figure 5, when using the same scale of the filtering kernel, the FDIF implements similar light estimation results to the Gaussian filter, which means our FDIF is effective in the light-estimation aspect. Moreover, the FDIF version of light estimation preserves edges better, while the Gaussian filter blurs them and therefore causes "over-smoothing" on the estimation image.

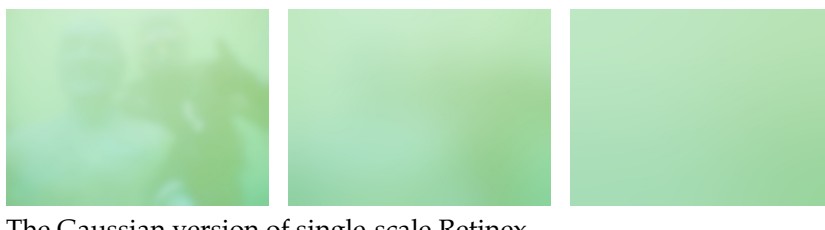

The Gaussian version of single-scale Retinex

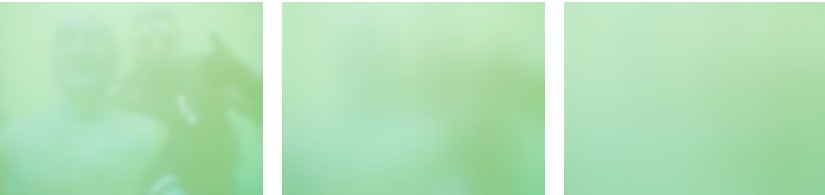

FDIF version of single scale Retinex when $\alpha = 0.5$

**Figure 5.** Comparison between the Gaussian and FDIF-based light estimation, *kernel_size* = $45, 243, 753$ from left to the right. The kernel size is chosen to be as close to the original MSR framework as possible to ensure accuracy, while the size must be odd for FDIF. In the first column, we can see that the FDIF version has kept the texture of the metal connector at the end of the breathing hoses, while the edges could barely be seen in the Gaussian version.

### 3.2.3. Color Restoration

Since underwater images often present disastrous color distortion, a color restoration method is decisive to the algorithm's overall performance. One of the classical color restoration approaches, gray-world white balancing, shows stability and validity in underwater color restoration tasks [20]. However, the Retinex processing of images sometimes brings about local or global violations for the gray-world theory, which assumes that the average intensities for R, G, and B channels tend to be constant. Physically, the assumption supposes the average reflection of light from natural objects is a constant. A logarithmic color restoration function (CRF) is therefore proposed by Jobson [9]:

$$C_i(x,y) = \beta \log[\alpha I_i'(x,y)] = \beta\{\log[\alpha I_i(x,y)] - \log[\sum_{i=1}^{S} I_i(x,y)]\} \tag{20}$$

where $C_i(x,y)$ denotes the CRF for channel $i$, $I_i(x,y)$ denotes the intensity on channel $i$, $S = 3$ representsan RGB image, and $\alpha = 125$ and $\beta = 46$ are empirical parameters for the underwater scene. In fact, $I_i'(x,y)$ represents a gray-world white balancing process. Different values of empirical parameters $\alpha$ and $\beta$ were also tested in Figure 6, and the chosen values are believed to be optimal.

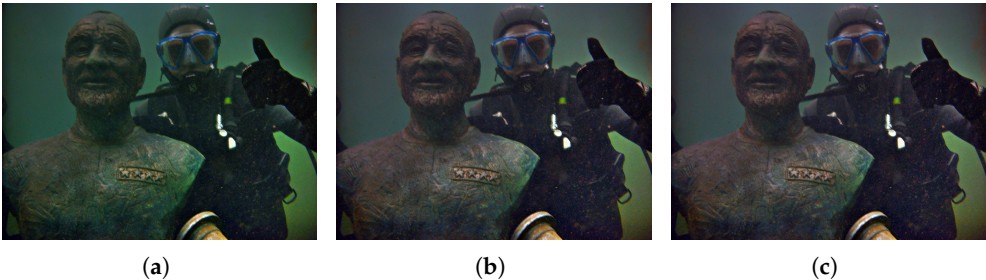

|            (a)            |            (b)            |            (c)            |

**Figure 6.** Empirical parameters for (**a**): $\alpha = 20$, $\beta = 20$; (**b**): $\alpha = 125$, $\beta = 46$; (**c**): $\alpha = 219$, $\beta = 69$. Compared to the optimal parameters in (**b**), small values for (**a**) brought about unsatisfactory correction for color distortion, while large values for (**c**) resulted in gray-out.

As an image enhancement method based on the domain transform, the Retinex enhanced the image by converting the pixel matrix to the logarithmic domain and reducing noise. After the Retinex process, an inverse transformation called quantization, which converts the continuous logarithmic pixels back to RGB space, should be imposed, and the process determines the performance to a large extent. Due to the wide dynamic range of the logarithmic images, most Retinex-based frameworks apply a canonical gain method for inverse transform instead of linear quantization:

$$R_{out}(x,y) = G \times C_i(x,y) \times \log R(x,y) + b \tag{21}$$

Here, $G$ and $b$ are empirical parameters, which are chosen to be 192 and $-30$, and $R_{out}(x,y)$ presents the final output of the MSRCR. However, for underwater images, this canonical gain measure results in gray-out, which means the output image appears to turn gray and lacks fresh colors. To refine the algorithm, Parthasarathy's method [21] has been proven to be effective for underwater images. By clipping the largest and smallest pixels of the logarithmic image to fixed values, the highly deviated maximum and minimum values are excluded from the quantization process. Then, the linear quantization method is used to transform the other values to RGB space, and the excluded largest and smallest parts of the pixels are set to 255 and 0, respectively. Furthermore, we utilize a gamma correction to the transformed pixel matrix, and compared to Parthasarathy's method, ours accomplished better color rendition.

### 3.3. Unsupervised Encoder–Decoder Network

After the Retinex with FDIF preliminarily enhanced the input underwater image, an unsupervised autoencoder network was employed to improve the image quality. For the impossibility of obtaining both raw data of an underwater image and its ground truth simultaneously in the real-world environment, the network is designed to be trained in an unsupervised manner.

#### 3.3.1. Network Architecture

Based on the prevailing U-Net [22] architecture, our encoder–decoder network takes a pre-enhanced underwater image as its input and generates a further improved output image. As depicted in Figure 7, the model is composed of the basic U-Net structure and attention modules. Four loss functions were taken into account so the training could converge.

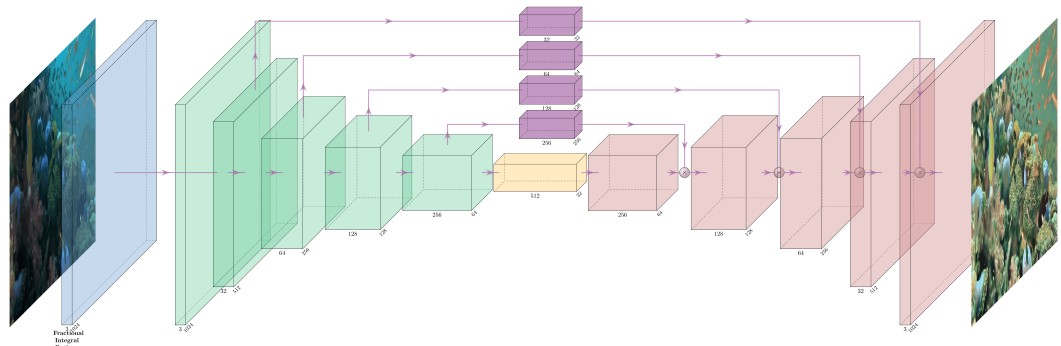

**Figure 7.** The structure of the proposed end-to-end underwater-image-enhancement framework.

First, the proposed framework takes an underwater image as input and applies the FDIF-MSR to the image. Then, the enhanced image will be refined by the unsupervised encoder–decoder network to alleviate noise brought in by the Retinex algorithm and improve detailed performance. The overall pipeline is shown in Figure 7.

#### 3.3.2. Attention Module

In this paper, a squeeze-and-excitation module [23] first proposed by Jie Hu et al. was integrated into our network. By squeezing each input feature channel into a descriptor, the SE-Block concentrates on exploiting channel-wise dependency and therefore expresses the whole image. The excitation operator maps the descriptors into channel weights to extend conventional local receptive fields to a representation for a cross-channel fusion of features, which can be regarded as a self-attention function.

#### 3.3.3. Loss Function

As for unsupervised learning, the loss function largely determines whether the neural network can effectively fit the data features. Based on the proposed encoder–decoder network, we introduce four losses to guide the training process.

- *Color Loss.* To mitigate the color difference between pre-enhanced and post-enhanced underwater images, color loss was introduced into our model. The color loss function is defined by the angle between the input and output pixel vectors:

$$L_{color} = \sum_{i,j} \angle (I_{i,j}, F(I_{i,j})) \tag{22}$$

where $F$ denotes a pixel matrix transform, and $I_{i,j}$ represents a single pixel in the matrix. As $L_2$ distance is widely used in image-processing tasks, a disadvantage has shown that the $L_2$ norm only calculates the numerical difference between the pixels,

but the directional difference of the pixel vectors cannot be measured. In the proposed model, the color loss aims to narrow the gap of the angle between pixel vectors but not introduce too much computational cost.

- *Mix-$L_1$-SSIM Loss.* Since the network is designed to learn to produce visually pleasing images, it is natural that a perceptually motivated loss function should be adopted in the training pipeline. Structural Similarity, also known as SSIM, is defined as:

$$SSIM(I_x, I_y) = \frac{2\mu_x\mu_y + C_1}{\mu_x^2 + \mu_y^2 + C_1} \cdot \frac{2\sigma_{xy} + C_2}{\sigma_x^2 + \sigma_y^2 + C_2} \tag{23}$$

where $\mu_x$ and $\sigma_x$ denote mean and variance of pixel matrix $x$, respectively. Constants $C_1$ and $C_2$ are determined by the dynamic range of the pixel. Consequently, the loss function of SSIM can be written as:

$$L_{SSIM} = 1 - SSIM(I_{input}, I_{output}) \tag{24}$$

By fine-tuning constants in Equation (23), multi-scale SSIM can be expressed by

$$L_{MS\_SSIM} = 1 - MS\_SSIM(I_{input}, I_{output}) \tag{25}$$

where the multi-scale SSIM is defined as:

$$MS\_SSIM(I_x, I_y) = \prod_{m=1}^{M} \left(\frac{2\mu_x\mu_y + C_1}{\mu_x^2 + \mu_y^2 + C_1}\right)^{\beta_m} \cdot \left(\frac{2\sigma_{xy} + C_2}{\sigma_x^2 + \sigma_y^2 + C_2}\right)^{\gamma_m} \tag{26}$$

According to Zhou Wang et al. [24], the MS-SSIM excels at preserving the contrast in high-frequency areas and supplies more flexibility than SSIM in incorporating the variations of viewing conditions. However, shifting of colors can be introduced by MS-SSIM loss, which may result in monotonous color rendition. To achieve better color and luminance performance, the $L_1$ loss, which aims to maintain color and luminance stability, is combined with the MS-SSIM loss, and the *Mix-$L_1$-SSIM Loss* can be written as

$$L_{Mix\_L_1\_SSIM} = \alpha L_{MS\_SSIM} + (1 - \alpha)L_1 \tag{27}$$

where empirical parameter $\alpha$ is chosen to be 0.7 in our model.

- *Perceptual Loss.* First proposed by Justin Johnson et al., perceptual loss has been proved to be valuable by numerous unsupervised models on image super-resolution and style transfer tasks. Instead of $L_1$ or $L_2$ loss, which exactly matches pixels of target image $\hat{y}$ with input $y$, the perceptual loss encourages $\hat{y}$ to have a similar feature representation to $y$, which can be regarded as constraining semantic changes during the image-enhancement process. The feature reconstruction loss can be defined as:

$$L_{perceptual}(\hat{y}, y) = \frac{1}{C_j \times H_j \times W_j} \|\phi_j(\hat{y}) - \phi_j(y)\|_2^2 \tag{28}$$

where $C_j$, $H_j$, and $W_j$ represent the channel, height, and width of the feature map, respectively, and $\phi_j$ denotes a feature extraction operator. We utilize a VGG-19 pretrained model to extract features of multiple layers from the image $\hat{y}$ and $y$ and then calculate the Euclidean distance between them to measure the difference. By minimizing feature-reconstruction perceptual loss, the model is able to produce visually indistinguishable output image $\hat{y}$ from $y$.

- *Total Variation Loss.* To prevent over-fitting and encourage the model to have better generalization capability, we use total variation loss in addition. In a two-dimensional continuous framework, the total variation loss is defined by:

$$L_{TV} = \int_{\Omega} \sqrt{u_x^2 + u_y^2} dx dy \tag{29}$$

where $u_x = \frac{\partial u}{\partial x}$, $u_y = \frac{\partial u}{\partial y}$ and $u$ denotes the image, $x, y \in \Omega$. As $u_x$ represents derivative along the x-axis, by minimizing $u_x$, we can constrain the luminance difference between two adjacent pixels, and therefore, the overall noise of the output image can be suppressed.

## 4. Experiments

Elaborate experiments were conducted on public underwater image datasets, and both qualitative methods and quantitative methods were used for evaluating our algorithm. The datasets we used in this paper and some implementation details should be explained before we show the performance of our framework.

### 4.1. Datasets and Implementation Details

Three public underwater image datasets were used in our paper: Underwater Image Enhancement Benchmark (UIEB) [14], which includes 890 raw underwater images, Ocean-Dark [25] for low-light underwater image enhancement, including 183 images of low-light or unbalanced-light condition, and Stereo Underwater Image Dataset [26] by Katherine A. Skinner et al.

Firstly, to train our unsupervised model, we inspected the entire UIEB and subjectively picked 782 high-quality samples from the 890 images as our training set, which contains about 88% data of the original UIEB. The training set was expanded to four times that of the original selected images by simply rotating the images 90°, 180°, and 270°.

Secondly, the FDIF-MSR was applied to the pre-processed dataset to attain preliminary enhanced images. Since most parameters of the FDIF-MSR algorithm comprising fractional orders are determined empirically and independent of the subsequent encoder–decoder network, we saved the preliminary enhanced images from the training set after the FDIF-MSR parameters were decided and the parameters were believed to be optimal.

Next, for one particular unsupervised training process, we utilized the saved enhanced images as input and fine-tuned the network hyper-parameters. All alterable hyper-parameters in our framework are shown in Table 1.

**Table 1.** Alterable hyper-parameters in the proposed model.

| Hyper-Parameter | Value |
| :---: | :---: |
| training epoch | 200 |
| initial learning rate | 0.01 |
| $\lambda_{TV}$ | 0.5 |
| $\lambda_{color}$ | 0.2 |
| $\lambda_{perceptual}$ | 1.0 |
| $\lambda_{mix}$ | 0.4 |
| $\alpha_{frac}$ | 0.75 |
| $\sigma$ | 45,81,251 |
| $\alpha$ | 125 |
| $\beta$ | 46 |
| high clip | 0.01 |
| low clip | 0.01 |
| $\gamma$ | 0.7 |

The unsupervised encoder–decoder network was trained on a single RTX-3090 GPU, with two Xeon Silver 4210 CPUs and 128 GB memory. We implemented the algorithm in Python 3.8.12 and used PyTorch framework version 1.11.0.

### 4.2. Evaluation

The evaluation of image enhancement has always been a significant challenge in that the human visual system is quite different from that of machines, and it is of great difficulty to measure what a visually pleasing image is by discrete calculation. To tackle this issue,

we tried to evaluate our model in both qualitative and quantitative ways, and we obtained the conclusion that it achieves excellent performance on the validation set.

Qualitative Evaluation

As mentioned above, underwater images face two major challenges, one that light scattering and absorption cause poor visibility, and another that various attenuation characteristics of different frequencies of light result in terrible color distortion. In Figure 8, and Figure 9, our algorithm is tested on both hazy and color distortion examples.

Such hazy images in Figure 8 are common in real underwater images taken by unmanned undersea vehicles (UUV), and our framework achieved remarkable results on these images in that heavy haze brought in by light scattering and absorption was removed while unnatural color deviation was mitigated as well.

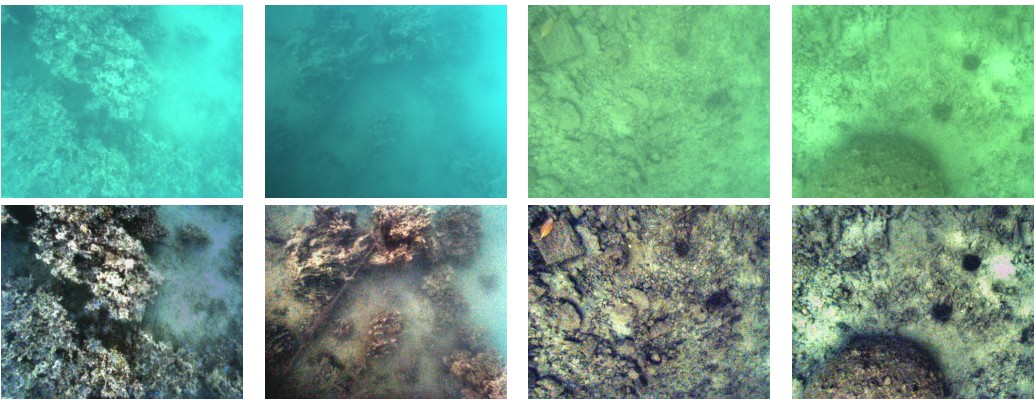

**Figure 8.** The hazy examples were chosen from the Stereo Underwater Image Dataset [26]. Although the dataset provided paired images taken by stereo cameras, we treat the images as if they were taken by monocular cameras in our work. **Top** row: hazy images; **Bottom** row: corresponding enhanced images.

In the first and second columns of Figure 9, the entire images suffered from color casts of blue and green, respectively, but our framework has effectively corrected the distortion. As for the third and fourth columns, the framework mainly focuses on eliminating color distortion of the foreground object in the images; thereby, the divers' and their equipment's natural color has been recovered observably.

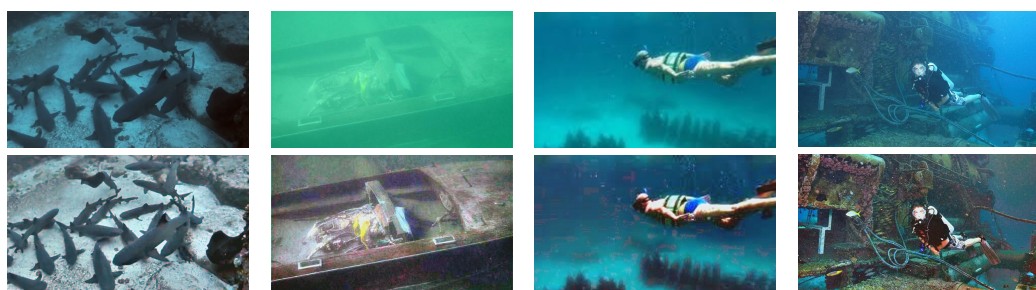

**Figure 9.** Some color distortion examples from the UIEB dataset. **Top** row: images with color casts; **Bottom** row: corresponding enhanced images.

Furthermore, as most underwater images were photographed under extremely low-light conditions, we tested our model for that as well, and some of the results are shown in Figure 10.

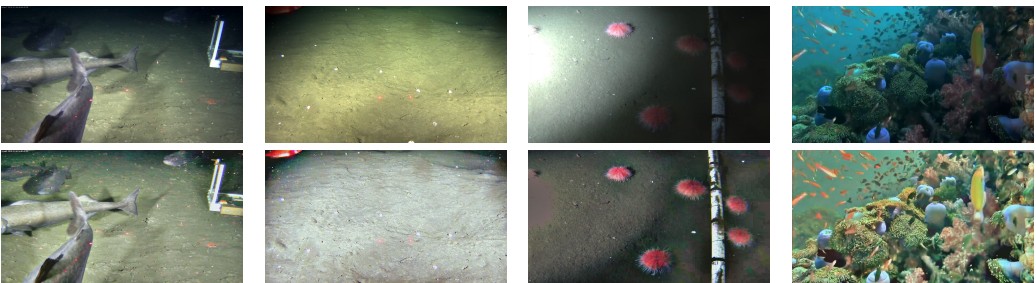

**Figure 10.** Three low-light examples on the left were chosen from the OceanDark dataset, and the other one was from the UIEB. **Top** row: images with unbalanced light; **Bottom** row: corresponding enhanced images.

In the case of the low-light condition, our framework achieves terrific results in that details in dark areas were enhanced vastly, while objects in bright areas were not over-enhanced and retained normal brightness.

As shown in Figure 11, we selected some images from the three above-mentioned public datasets for conducting the visual comparison. The chosen images were taken from diverse and challenging underwater scenes that include underwater haze, distortion of color, and low-light conditions. One can observe that our algorithm presented high robustness and satisfactory results under various underwater environments. Compared to the state-of-the-art method, TACL, our model shows equal effectiveness in color correction and low-light enhancement and performs with better haze-removal results (the sixth row) with clear details and sharp structures. Specifically, among the third row of the images, Chen's method did not enhance the input image much; the UWCNN blurred objects in the shadows; the FUnIE-GAN brought undesirable color distortion; and the TACL failed to balance objects under different lighting conditions. However, our model has successfully overcome these problems.

### 4.3. Quantitative Evaluation

To make our analysis more convincing, some of the widely used full-reference image quality evaluation metrics were applied to our algorithm. Firstly, we considered the PSNR metric. The PSNR, also known as peak signal-to-noise ratio, is defined as

$$PSNR = 10 \times log_{10}(\frac{255^2}{MSE}) \tag{30}$$

where the *MSE* represents the pixel-wise difference between the input and output images by the expression

$$MSE = \frac{1}{MN} \sum_{i=1}^{M} \sum_{j=1}^{N} (I'_{ij} - I_{ij})^2 \tag{31}$$

where *M* and *N* represent the height and width of the image, respectively, and $I'_{ij}$ indicates a specific pixel at $(i, j)$.

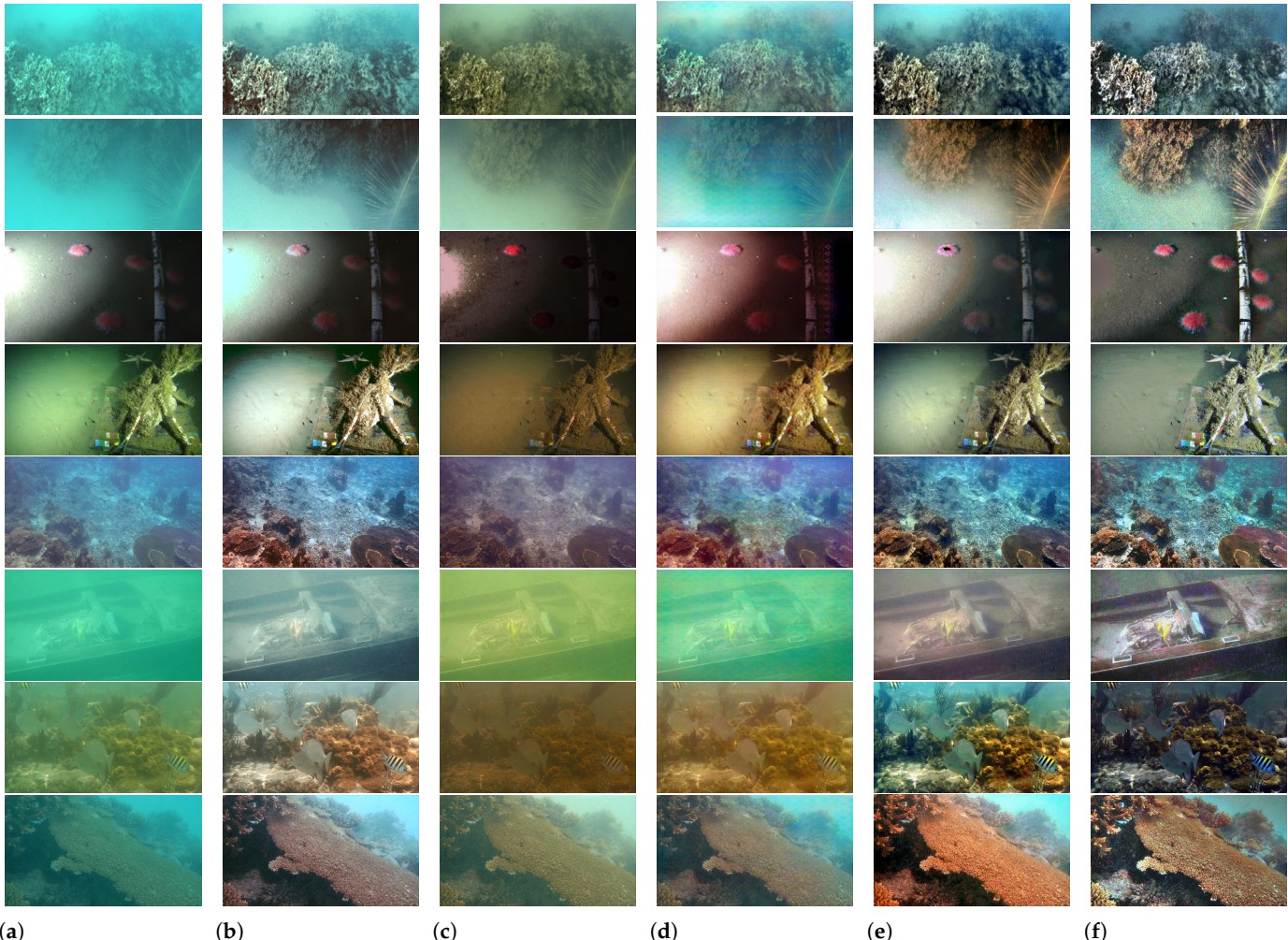

| (**a**) | (**b**) | (**c**) | (**d**) | (**e**) | (**f**) |

**Figure 11.** Images for qualitative comparison between our method and others, including a method based on deep learning and image formation model by Chen et al., UWCNN, FUnIE-GAN and TACL. (**a**) Input image. (**b**) Chen et al. [27]. (**c**) UWCNN [28]. (**d**) FUnIE-GAN [29]. (**e**) TACL [17]. (**f**) Ours.

Secondly, we utilize the SSIM metric, which was defined above by Equation (23). The SSIM is based on the hypothesis that the human's visual system excels at perceiving structural information from real scenes, and hence the structural similarity can be an appropriate approximation to the image quality. Furthermore, two significant image properties, contrast and luminance, were also considered in our evaluation process.

Since not all of the images in the datasets we used are of high quality in terms of both color and texture, and quite a few images from the datasets were monotonous in the scene and tonality, we only show the evaluation results on a subset of the typical and high-quality samples instead of evaluating on the entire datasets. Each image we subjectively selected for the evaluation is based on the fact that most of the underwater images that required an enhancement were taken by underwater vehicles or divers in areas of poor visibility, so we specifically excluded those photographed near the surface of the water with strong natural uniform light.

The evaluation results are shown in Figure 12 and Table 2.

**Table 2.** Quantitative evaluation results on the datasets.

| Image Number | PSNR | CER [1] | LER | SSIM |
|:---:|:---:|:---:|:---:|:---:|
| UIEB-65 | 27.624 | 0.827 | 5.471 | 0.692 |
| UIEB-66 | 27.627 | 0.924 | 3.653 | 0.772 |
| UIEB-99 | 28.125 | 2.192 | 1.451 | 0.692 |
| UIEB-416 | 27.756 | 1.626 | 1.732 | 0.568 |
| UIEB-715 | 27.809 | −0.607 | 1.195 | 0.803 |
| HIMB-1 | 27.819 | 2.004 | 1.192 | 0.632 |
| HIMB-2 | 27.833 | 3.081 | 1.309 | 0.427 |
| HIMB-3 | 27.751 | 1.877 | 1.255 | 0.435 |
| HIMB-4 | 27.946 | 0.701 | 1.150 | 0.890 |
| HIMB-5 | 27.797 | 2.728 | 1.092 | 0.625 |
| OceanDark-2 | 27.768 | −0.358 | 0.839 | 0.899 |
| OceanDark-5 | 27.929 | −0.663 | 1.244 | 0.910 |
| OceanDark-145 | 27.969 | −0.267 | 0.776 | 0.905 |
| OceanDark-155 | 27.666 | −0.287 | 1.778 | 0.797 |
| OceanDark-164 | 27.703 | −0.232 | 1.771 | 0.803 |

[1] CER and LER in Equations (32) and (33) refer to contrast enhancement rate and luminance enhancement rate, respectively. They can be calculated by the contrast and luminance difference between input and output images, respectively. The values are normalized by the input.

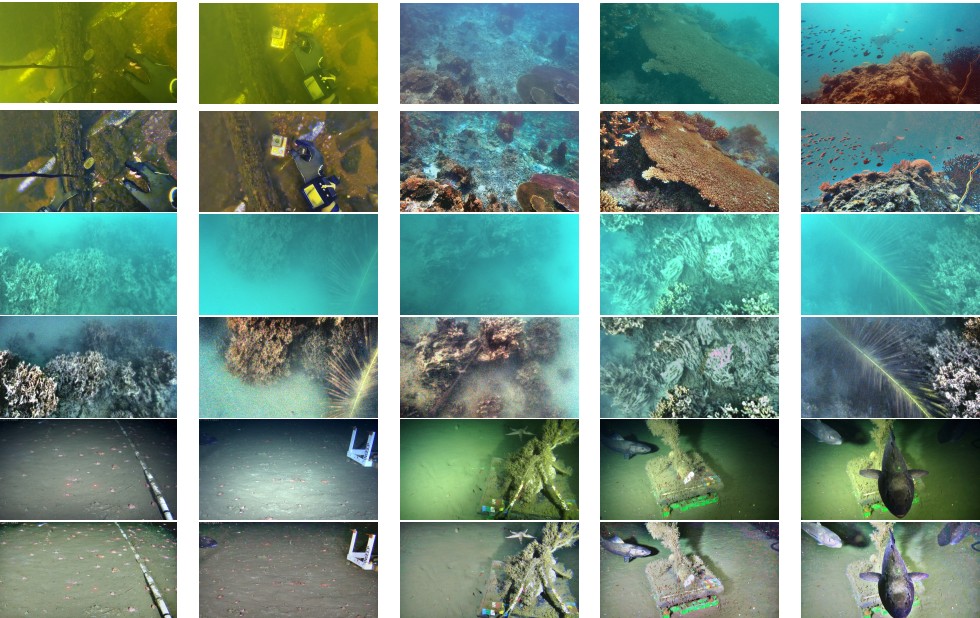

**Figure 12.** Images for quantitative evaluation. **Top** two rows: UIEB-65, UIEB-66, UIEB-99, UIEB-416, UIEB-715 from the UIEB dataset and corresponding post-enhancement images; **Middle** two rows: images numbered HIMB-1 to HIMB-5 from the Stereo Underwater Image Dataset and their corresponding post-enhancement images; **Bottom** two rows: OceanDark-2, OceanDark-5, OceanDark-145, OceanDark-155, OceanDark-164 from the OceanDark dataset and corresponding post-enhancement images.

$$CER = \frac{1}{C} \sum_{c=1}^{C} Var(I_c) \tag{32}$$

where $C$ denotes the number of channels and $Var(I_c)$ denotes the variance of the image luminance on channel $c$.

$$LER = \frac{1}{MN} \sum_{i=1}^{M} \sum_{j=1}^{N} \frac{I_{out}(i,j) - I_{in}(i,j)}{I_{in}(i,j)} \tag{33}$$

As shown in Figure 11, the proposed model performed better than many other popular methods on underwater images. In Table 3, we provide the average PSNR, LER, CER, and SSIM values over the evaluation set used in Figure 11 for comparing the results of our model and the state-of-the-art methods. Since the four metrics are independent and lack a uniform representation of the enhancement performance, to quantitatively evaluate the image enhancement performances better, here, we introduced a uniform overall score that takes all four impact factors into account. The score is calculated by:

$$Score = \alpha \times PSNR + \beta \times |LER - CER|^2 + \gamma \times SSIM \tag{34}$$

where weight coefficients $\alpha = 0.6$, $\beta = -0.5$, and $\gamma = 2$ were selected so the overall score could better represent the image enhancement performance. In Equation (34), we designed $|LER - CER|^2$ to be a penalty term because unmatched luminance and contrast enhancement harm the visual quality of the image such that the images may have good results on quantitative values but fail on the human perceptual system. The results indicate that our model has similar LER and CER performance to the TACL but higher PSNR and SSIM and that our model presented even better results than TACL, the SOTA. Although the UWCNN and FUnIE-GAN yield better luminance enhancement rates, the two methods have bad results in terms of contrast enhancement, so they result in less pleasing enhanced images, as shown in Figure 11. Among all five methods, our model yielded the best overall score and presented visually pleasing results.

**Table 3.** Quantitative comparison between the proposed model and other methods in Figure 11.

| Model | PSNR | LER | CER | SSIM | Score |
|---|---|---|---|---|---|
| Chen et al. | 27.281 | 1.699 | 1.965 | 0.825 | 17.983 |
| UWCNN | 27.026 | 2.026 | −0.105 | 0.784 | 15.513 |
| FUnIE-GAN | 27.313 | 2.029 | 0.698 | 0.777 | 17.056 |
| TACL | 27.246 | 1.590 | 1.986 | 0.770 | 17.809 |
| **Ours** | 27.802 | 1.583 | 1.955 | 0.849 | 18.310 |
| Best | 27.802 (**Ours**) | 2.029 (FUnIE-GAN) | 1.986 (TACL) | 0.849 (**Ours**) | 18.310 (**Ours**) |

To make our paper more convincing, an ablation study was imposed on the proposed model. Models were evaluated on the dataset we used in Figure 12. Experiments were organized as follows:

(1) **Model No. 1**: No encoder–decoder network is used for refining the result of the proposed FDIF-Retinex;

(2) **Model No. 2 to 5**: The network is trained by the specific combination of loss function;

(3) **Model No. 6**: The SE-Block is replaced with direct residual connections.

(4) **Model No. 7**: The proposed full model.

As shown in Table 4, the FDIF-Retinex without post-enhancement (No. 1) has higher PSNR, LER, and CER but lower SSIM and overall score compared to the full model. This indicates that a post-enhancement network benefits the enhancement process by suppressing over-enhancement, which brings about more visually pleasing results. Through the results of the ablation study, we can see that if the model is trained by TV loss only or color loss with perceptual loss, the models perform poorly on the evaluation set and have a negative impact on the contrast enhancement rate. The models trained by perceptual loss only or perceptual loss and mixed loss have better performance than the above, but we want more natural results to be achieved=. To train the network, by combining the four loss functions, we can achieve a balance between luminance and contrast enhancement performance so that the output images will not be too bright or unnatural. Considering replacing residual connections with the SE-Block, the proposed model yielded better results on contrast enhancement and remained equal luminance enhancement capability, SSIM, and a higher PSNR. According to the experiments' results, our method adopted an appropriate loss

function group for training and an influential network architecture to improve underwater image quality.

**Table 4.** Ablation study results on network architecture and loss functions [1].

| Model No. | TV Loss | Color Loss | Percep. Loss | Mix Loss | SE-Block | PSNR | LER | CER | SSIM | Score |
|---|---|---|---|---|---|---|---|---|---|---|
| 1 | - | - | - | - | - | 27.884 | 1.865 | 2.102 | 0.605 | 17.912 |
| 2 | ✓ | - | - | - | ✓ | 27.979 | 2.239 | −1.000 | 0.664 | 12.870 |
| 3 | - | ✓ | ✓ | - | ✓ | 27.824 | 3.385 | −0.390 | 0.574 | 10.717 |
| 4 | - | - | ✓ | - | ✓ | 27.859 | 1.638 | 1.126 | 0.634 | 17.852 |
| 5 | - | - | ✓ | ✓ | ✓ | 27.748 | 1.730 | 1.769 | 0.625 | 17.898 |
| 6 | ✓ | ✓ | ✓ | ✓ | - | 27.777 | 1.705 | 1.107 | 0.662 | 17.811 |
| 7 [2] | ✓ | ✓ | ✓ | ✓ | ✓ | 27.811 | 1.657 | 1.395 | 0.647 | **17.946** |

[1] The results are reported for the dataset we used in Figure 12. [2] The No. 7 model refers to enhancing the image by FDIF-Retinex only without a post-enhancement network.

## 5. Conclusions

In this paper, we proposed an end-to-end underwater-image-enhancement framework that excels at color restoration and haze removal for underwater scenes. Based on the fractional double integral filter, the proposed FDIF algorithm yielded better results on edge preservation than the widely used Gaussian version of multi-scale Retinex. An unsupervised encoder–decoder network that integrates an advanced attention mechanism and well-designed loss functions was utilized to further improve the quality of the enhanced images. Both qualitative evaluation and quantitative evaluation showed the effectiveness of the proposed framework, which achieved superb performance across multiple datasets and various underwater environments. In the future, we are planning to deploy the proposed underwater-image-enhancement model on embedded devices and test it in a real-world environment. We hope the model will provide a brand new view of underwater-images-enhancement methods, and we believe that the proposed model could benefit other downstream tasks such as object detection and 3D reconstruction.

**Author Contributions:** Conceptualization, Y.Y.; methodology, Y.Y. and C.Q.; software, C.Q.; validation, C.Q.; resources, Y.Y.; data curation, C.Q.; writing—original draft preparation, C.Q.; writing—review and editing, Y.Y.; visualization, C.Q. All authors have read and agreed to the published version of the manuscript.

**Funding:** This work was supported by National Key Research and Development Program [grant numbers 2021YFC2803000, 2020FYB1313200]; the National Natural Science Foundation of China [grant numbers 52001260], and the National Basic Scientific Research Program [grant numbers JCKY2019207A019].

**Institutional Review Board Statement:** Not applicable.

**Informed Consent Statement:** Not applicable.

**Data Availability Statement:** The data presented in this study are openly available in https://li-chongyi.github.io/proj_benchmark.html for the UIEB (accessed on 14 April 2022), https://sites.google.com/view/oceandark/home for the OceanDark (accessed on 12 September 2022), https://github.com/dlut-dimt/Realworld-Underwater-Image-Enhancement-RUIE-Benchmark for the RUIE (accessed on 13 December 2021), and https://github.com/kskin/data for the Stereo Underwater Image Dataset (accessed on 28 June 2022).

**Conflicts of Interest:** The authors declare no conflicts of interest.

## Abbreviations

The following abbreviations are used in this manuscript:

| | |
|---|---|
| MSR | Multi-Scale Retinex |
| MSRCR | Multi-Scale Retinex with Color Restoration |
| CNN | Convolutional Neural Network |
| GAN | Generative Adversarial Network |
| FDIF | Fractional Double Integral Filter |
| SSIM | Structural Similarity |
| TV | Total Variation |
| UUV | Unmanned Underwater Vehicle |
| PSNR | Peak Signal-to-Noise Ratio |
| MSE | Mean Square Error |
| LER | Luminance Enhancement Rate |
| CER | Contrast Enhancement Rate |

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
