# Peer review of "An End-to-End Underwater-Image-Enhancement Framework Based on Fractional Integral Retinex and Unsupervised Autoencoder"

_fractalfract, doi:10.3390/fractalfract7010070_

Round 1

Reviewer 1 Report

An End-to-End Underwater Image Enhancement Framework Based On Fractional Integral Retinex and Unsupervised Autoencoder

Comments

1.    Introduction

Line 97: ....mostly used

Lines 110-112: An effective unsupervised encoder-decoder network, requiring no adversarial training and yielding perceptually pleasing results, was employed to refine the output of the previous Retinex model.

Line 115: .....produced impressive results.

Line 117: ...... of fractional double integral, on which the proposed variant of Retinex is based, ….

Note: More existing works on underwater image restoration algorithms should be cited and discussed.

2.    Mathematical Background

Line 136: There is an equivalent relation……..

Line 143: Note that……

Lines 136-138: There is an equivalence relation between Caputo and R-L derivatives that for a positive real number α, satisfied 0 <= n − 1 < α < n. If function f (t) defined on the interval [a, b] has continuous derivatives of n − 1 order and integrable, we have

Line 146: …….operator theory, we use

Line 148: instead of α. For G-L definition, ……

Note: What mathematical models exist for the formation of underwater degraded images? This will provide a good background as to how underwater degraded images are formed. Like the mathematical model for the formation of hazy images in outdoor scenes as seen in [1, 2, 3, 4].

3.    Proposed Method

Line 165: ……a more simplified version…..

Line 167: Generalizing this simplified equation……

Line 167: which involving along the x-axis positive…… *Reconstruct this phrase

Line 182: …..the Equation (11) becomes

Line 209: In Retinex theory, …….. (A new paragraph should start from here)

Line 255: the pixels are set to 255…..

Line 260: …..will be employed to improve the image quality better.

4.    Experiments

Line 364: Such hazy images in Figure 7 are common in real underwater images……..

Line 368: In the first and second columns of Figure 8, the entire images suffered from color……

Line 380: To make our analysis more convincing,……..

Line 382: The PSNR, also known as peak signal-to-noise ratio, is defined by:

Line 386: Secondly, we utilize the SSIM…. (A new paragraph should start from here)

*Note: (1) Present the mathematical definitions for computing the CER and LER.

(2) Please, compare your results against existing methods.

Ref.

[1] S.G. Narasimhan and S.K. Nayar, “Vision and the Atmosphere, Int’l J. Computer Vision, vol. 48, pp. 233-254, 2002.

[2] S.G. Narasimhan and S.K. Nayar, “Chromatic Framework for Vision in Bad Weather,” Proc. IEEE Conf. Computer Vision and Pattern Recognition, vol. 1, pp. 598-605, June 2000.

[3] R. Fattal, “Single Image Dehazing,” Proc. ACM SIGGRAPH ’08, 2008.

[4] R. Tan, “Visibility in Bad Weather from a Single Image,” Proc. IEEE Conf. Computer Vision and Pattern Recognition, June 2008.

Author Response

Response to Reviewer 1 Comments

Point 1: Language modification.

Response 1: All changes suggested by the reviewer have been completed. We also made some modifications to other inappropriate words.

Point 2: More existing works on underwater image restoration algorithms should be cited and discussed.

Response 2: We added a state-of-the-art underwater image enhancement method, the TACL, in our introduction part, and compared it with our method later.

Point 3: What mathematical models exist for the formation of underwater degraded images? This will provide a good background as to how underwater degraded images are formed. Like the mathematical model for the formation of hazy images in outdoor scenes as seen in [1, 2, 3, 4] (References provided by the reviewer).

Response 3: We looked into the papers provided by the reviewer, and we found some things in common between underwater image enhancement and image dehazing on land. We have added corresponding supplementary content and references to our paper.

Point 4: Present the mathematical definitions for computing the CER and LER.

Response 4: We added the definitions for computing the CER and LER in Equation(32) and Equation(33), respectively.

Point 5: Please, compare your results against existing methods.

Response 5: We added qualitative comparison results and ablation study results in 4.2.

Reviewer 2 Report

1. Authors should present the proposed algorithm in the form of flow chart or block diagram.

2. The comparison of the proposed work is not done with the existing methods.

3. Recent research work needs to be included in the proposed work.

4. The robustness of the proposed work should also be discussed in the adverse conditions.

Author Response

Response to Reviewer 2 Comments

Point 1: Authors should present the proposed algorithm in the form of flow chart or block diagram.

Response 1: A block diagram of the proposed algorithm is added and as shown in Fig.1.

Point 2: The comparison of the proposed work is not done with the existing methods.

Response 2: We added qualitative comparison results between our method and other existing methods and ablation study results in 4.2.

Point 3: Recent research work needs to be included in the proposed work.

Response 3: We added a state-of-the-art underwater image enhancement method, the TACL, in our introduction part, and compared it with our method later. More recent research works have been included in our original version of the manuscript in Paragraphs 4 and 5 in Section 1.

Point 4: The robustness of the proposed work should also be discussed in the adverse conditions.

Response 4: As shown in Fig.11(added), the images for evaluation are taken from diverse and challenging underwater scenes, including underwater haze, distortion of color, and low-light conditions. We discussed the robustness of our algorithm at the end of 4.2.

Reviewer 3 Report

This paper proposed a method for underwater image enhancement. More specifically, the authors proposed an integral-based Retinex to preserve more edges in the input image. An unsupervised training strategy was proposed to address the concern of lack of enough ground truth data in underwater situations. Results on three datasets were reported. 

=========

Overall, this paper is easy to understand, but my major concern is on the experiment section:

- The author may want to compare the proposed method with other state-of-the-art methods to verify the effectiveness of the proposed method on the three dataset used in this paper. Based on current results, I was not convinced by the effectiveness of the proposed method. 

- The authors may want to report comparison results to validate the proposed fractional double integral by comparing the baseline with fractional integral.

- An U-net with squeeze-and-excitation module was used in the paper, and the authors may want to add an ablation study on the network architecture. 

- Four different losses were proposed for unsupervised learning. Do any of them really improve the performance?

Author Response

Response to Reviewer 3 Comments

Point 1: The author may want to compare the proposed method with other state-of-the-art methods to verify the effectiveness of the proposed method on the three dataset used in this paper. Based on current results, I was not convinced by the effectiveness of the proposed method. 

Response 1: We added qualitative comparison results between our method and other existing methods in 4.2. Other methods include UWCNN, FUnIE-GAN, and TACL. We believe the TACL to be a state-of-the-art method with high robustness for various underwater environments and good performance for degraded images. Compared to the TACL, our model shows equal effectiveness in color correction and low-light enhancement and performs better haze removal results with clear details and sharp structures.

Point 2: The authors may want to report comparison results to validate the proposed fractional double integral by comparing the baseline with fractional integral.

Response 2: As shown in Fig.3 (modified), we conducted a visual comparison between the proposed fractional double integral and the fractional integral. The latter shows apparent unpleased artifacts, and the filtered image is not acceptable in visual performance.

Point 3:

1)An U-net with squeeze-and-excitation module was used in the paper, and the authors may want to add an ablation study on the network architecture. 

2)Four different losses were proposed for unsupervised learning. Do any of them really improve the performance?

Response 3: We showed the results of an ablation study on the network architecture and the four different losses proposed for unsupervised learning, and discussed them at the end of Section 4. Note that if only color loss or the mix loss is used, the model parameters are difficult to converge, so we combined the perceptual loss with them to help the training process.

Round 2

Reviewer 2 Report

Authors have incorporated all the suggestions that improved the quality of the manucript.

Author Response

More experiments for comparing the proposed method with other methods have been done and added in Section 4.

Reviewer 3 Report

Thanks for the authors’ responses, and they partially addressed my concerns.

===========

- The image quality in this version is way worse than before. I am not able to tell the visual comparison in all figures. In particular, the newly added figure 3.

- I still have concerns about the effectiveness of the proposed method. I appreciate the qualitative results reported in fig 11, but I would like to see more quantitative comparisons, such as a table, to justify the proposed method. 

A sample comparison table can be found in TABLE I, Fast Underwater Image Enhancement for Improved Visual Perception.

- The ablation results reported in table 3 were not clear, the authors may want to make a better table and add more description and discussion.

A sample ablation study table can be found at TABLE IV, DispSegNet: Leveraging Semantics for End-to-End Learning of Disparity Estimation from Stereo Imagery.

- The authors may want to highlight the improvement in fig 11 and include a description of which dataset they reported on. 

- The authors may want to include a description of which dataset they reported on in table 2.

Author Response

Response to Reviewer 3 Comments

Point 1: The image quality in this version is way worse than before. I am not able to tell the visual comparison in all figures. In particular, the newly added figure 3.

Response 1: We added SSIM to quantitatively compare the FI version and the FDI version of the filtered images in Fig.3. The SSIM performs well in comparing images with artifacts and images with smooth structural details. The flaw of the FI filter is that it brought in extra texture that did not even exist in the input image.

Point 2: I still have concerns about the effectiveness of the proposed method. I appreciate the qualitative results reported in fig 11, but I would like to see more quantitative comparisons, such as a table, to justify the proposed method. 

Response 2: In addition to the qualitative comparison results shown in Fig.11, we added qualitative results in the new Table.3 and added more discussions.

Point 3: The ablation results reported in table 3 were not clear, the authors may want to make a better table and add more description and discussion.

Response 3: We have remade the table according to the reference material. More discussions were also added at the end of 4.3.

Point 4: The authors may want to highlight the improvement in fig 11 and include a description of which dataset they reported on. 

Response 4: 1) We added more descriptions in the newest version of the manuscript. 2) We added an explanation of which dataset the results reported on.

Point 5: The authors may want to include a description of which dataset they reported on in table 2.

Response 5: As shown in Table.2, we used the UIEB, HIMB, and OceanDark datasets. We added a description in the \caption{} of Figure.12.

Round 3

Reviewer 3 Report

Thanks for the authors’ responses, and they partially addressed my concerns.

===========

- The author may want to highlight the best performance for each column in table 3. It is not clear whether the proposed method outperformed other approaches.

- Similarly, highlight the best performance for each column in table 4.

- The discussion for table 4 is vague. For example, “As shown in Table 4, the FDIF-Retinex without post-enhancement has higher PSNR, LER, and CER but lower SSIM compared to the full model” there are many results in table 4, which two models did the authors compare?

- The authors may want to include a description of which dataset they reported on in table 3 and table 4.

Author Response

Response to Reviewer 3 Comments

Point 1:

(1) The author may want to highlight the best performance for each column in table 3. It is not clear whether the proposed method outperformed other approaches.

(2) Similarly, highlight the best performance for each column in table 4.

Response 1:

  • We highlighted the best performance for each column in Table.3. To make our proposed method more convincible, we reorganized descriptions for Table.3 and Table.4 in “4.3 Quantitative Evaluation”. We introduced an overall score for the uniform representation of the enhancement performance to quantitatively evaluate the image enhancement performances better. In Table.3, our method yields the highest score. In Table.4, the full model has the best performance.
  • The best value in the score column in Table.4 was emphasized.

Point 2: The discussion for table 4 is vague. For example, “As shown in Table 4, the FDIF-Retinex without post-enhancement has higher PSNR, LER, and CER but lower SSIM compared to the full model” there are many results in table 4, which two models did the authors compare?

Response 2: The sentence has been revised as required. From Line 440 to Line 443, we have listed all the models we experimented with for Table.4. The “FDIF-Retinex without post-enhancement” refers to model No.1.

Point 3: The authors may want to include a description of which dataset they reported on in table 3 and table 4.

Response 3:

Table.3. caption: “Quantitative comparison between the proposed model and other methods in Figure 11.”. In Line 384 and Line 385, we described where and how we acquired the evaluation set in Figure 11: “we selected some images from the above-mentioned three public datasets for conducting the visual comparison.”

Similarly, for Table.4, in Line 430: “Models are evaluated on the dataset we used in Figure 12.” We described where and how we acquired the evaluation set in the caption of Figure 12 in detail.